# Conventional and Real-Time PCR Targeting *bla*_OXA_ Genes as Reliable Methods for a Rapid Detection of Carbapenem-Resistant *Acinetobacter baumannii* Clinical Strains

**DOI:** 10.3390/antibiotics11040455

**Published:** 2022-03-28

**Authors:** Dagmara Depka, Agnieszka Mikucka, Tomasz Bogiel, Mateusz Rzepka, Patryk Zawadka, Eugenia Gospodarek-Komkowska

**Affiliations:** Microbiology Department, Ludwik Rydygier Collegium Medicum in Bydgoszcz, Nicolaus Copernicus University in Toruń, 85-094 Bydgoszcz, Poland; a.mikucka@cm.umk.pl (A.M.); bogiel.tomasz@wp.pl (T.B.); mateusz.rzepka@cm.umk.pl (M.R.); patrzawa@gmail.com (P.Z.); gospodareke@cm.umk.pl (E.G.-K.)

**Keywords:** *Acinetobacter baumannii*, carbapenemases, carbapenem-resistant, OXA-like beta-lactamases, OXA-like carbapenemases, OXA-23, OXA-40, resistance to carbapenems

## Abstract

Multidrug-resistant *Acinetobacter baumannii*, particularly those producing carbapenemases, are spread worldwide. A reliable detection of carbapenemases is essential to choose the appropriate antimicrobial therapy and, consequently, prevent the dissemination of carbapenem-resistant strains. The aim of this study is to examine the molecular basis of the carbapenem resistance mechanism and estimation of conventional PCR and real-time PCR usefulness for the detection of oxacillinases when compared to phenotypic carbapenemases detection. The following methods were evaluated: the CarbAcineto NP test, Carbapenem Inactivation Method, CPO panels of semiautomated antimicrobial susceptibility testing method on the BD Phoenix™ M50 system, conventional Polymerase Chain Reaction and real-time PCR. The eazyplex^®^ SuperBug complete A assay was used as the reference method. Among the tested strains, 39 (67.2%) carried the *bla*_OXA-40_ gene, while the *bla*_OXA-23_ gene was noted amongst 19 (32.8%) isolates. The diagnostic sensitivities of the studied assays were as follows: CarbAcineto NP—65.5%; CIM—100%; CPO—100%; conventional PCR—100%; real-time PCR—100%.

## 1. Introduction

*Acinetobacter baumannii* are Gram-negative nonfermenting rods, an etiological factor of opportunistic infections among hospitalized patients [1,2]. This species mostly causes respiratory and urinary tract infections and bacteriemia. The mentioned rods may also be involved in *meningitis*, bacterial *endocarditis* and eye, skin and soft tissue infections [3]. *A. baumannii* strains are often multidrug-resistant (MDR); therefore, the treatment of infections of this etiology is often quite problematic and frequently requires the inclusion of “salvage therapy” [4].

*A. baumannii* strains have a number of intrinsic resistance mechanisms, e.g., *Acinetobacter*-derived cephalosporinases (ADC) and OXA-51-like beta-lactamases. They are naturally resistant to ampicillin and amoxicillin and its combination with clavulanic acid, cefazolin, cefotaxime, cefriaxone, aztreonam, ertapenem, trimethoprim and fosfomycin [5].

Additionally, the *A. baumannii* genome may contain pathogenicity islands and plasmids, which determine the presence of genes encoding enzymes that hydrolyze different antibiotics, such as beta-lactamases (e.g., carbapenemases) [6]. Class A, B and D carbapenemases (according to Ambler’s classification) are the enzymes usually detected among these bacteria. The most prevalent amongst them are class D beta-lactamases, including Carbapenem-Hydrolyzing class D Beta-Lactamases (CHDLs and oxacillinases). The following oxacillinases groups have been previously identified among *A. baumannii* strains: OXA-51-like, OXA-23-like, OXA-40-like, OXA-58-like, OXA-143-like and OXA-235-like [7,8]. CHDLs usually present a weak hydrolytic activity of carbapenems, unless other resistance mechanisms are involved simultaneously, including efflux pumps, the loss of porins or other properties connected with the presence of insertion sequences (ISs) in their genome [8].

ISs have been detected upstream of *bla*_OXA_ genes. They serve as particular gene promoters, which lead to their overexpression [9]. ISs have the ability to relocate within the bacterial genome because the transposase enzyme gene can also be found in their structure. IS*Aba1* insertion sequences are the most commonly identified in *A*. *baumannii* genomes. It has been previously detected upstream of *bla*_OXA-23-like_, *bla*_OXA-40-like_ or *bla*_OXA-58-like_ genes [9]. In turn, *bla*_OXA-58-like_ genes occur more frequently within a different IS, called *Aba3* [10].

The reliable detection of carbapenemases is essential to prevent the spread of carbapenem-resistant strains and to choose the appropriate antimicrobial therapy [11]. Therefore, the aim of this study is to evaluate the molecular basis of the carbapenem resistance mechanism among *A. baumannii* strains and the evaluation of conventional PCR and real-time PCR usefulness for the detection of oxacillinases when compared to phenotypic detection methods. The following methods were evaluated: CarbAcineto NP, the Carbapenem Inactivation Method (CIM), CPO panels (Carbapenemase-Producing Organisms, CPO—GN-502 Panels, Becton Dickinson, Franklin Lakes, New Jersey, USA) of semiautomated antimicrobial susceptibility testing (AST) method on the BD Phoenix™ M50 system, and molecular biology-based approach: conventional Polymerase Chain Reaction (PCR), real-time PCR and eazyplex^®^ SuperBug complete A assay (AmplexDiagnostics GmbH, Gars am Inn, Germany) accompanied with the Genie^®^ II device (OptiGene). The latter, the in vitro diagnostic (IVD)-certified kit, was chosen to be dedicated to MDR Gram–negative rods and used as the reference method. It is a molecular assay based on the loop-mediated isothermal amplification (LAMP) real-time PCR method, which determines the presence of the most clinically important carbapenemases genes (KPC, NDM, VIM, OXA-48, OXA-23, OXA-40 and OXA-58) in bacteria cultures or directly in clinical specimen samples [12].

## 2. Results

### 2.1. Antimicrobial Susceptibility

*A. baumannii* strains presented a high percentage of resistance to all of the examined antimicrobials. All of the studied isolates (*n* = 58) were also resistant to both of the carbapenems. A total of 54 (93.1%) isolates was extensively drug-resistant (XDR), and 4 (6.9%) were pandrug-resistant (PDR). Most of the strains were susceptible to colistin only (91.4%; *n* = 53) (Table 1).

### 2.2. Results of the Reference Method of Carbapenemase Genes Detection

Among the examined strains, 39 (67.2%) carried the *bla*_OXA-40_ gene, while the *bla*_OXA-23_ gene was noted among 19 (32.8%) isolates using the eazyplex^®^ SuperBug complete A assay.

### 2.3. Results Comparison

Using CPO panels, the antimicrobial susceptibility profiles of all the studied strains were investigated, while classes of beta-lactamases were not determined for five (8.6%) of them (Table 2). Three of these isolates in the reference methods were *bla*_OXA-40_-positive and two were *bla*_OXA-23_-positive. Using the molecular biology-based approach, 100% (*n* = 58) of the strains were classified into a particular class of beta-lactamases.

The sensitivity of the studied assays is presented in Table 3. Among the 20 negative results obtained in the CarbAcineto NP test, 12 (60.0%) isolates were *bla*_OXA-23_-positive and 8 (40.0%) *bla*_OXA-40_-positive, with an application of the reference method.

In one isolate, the *bla*_OXA-23_ gene was detected using molecular methods (conventional PCR and real-time PCR), as opposed to the reference method, in which the *bla*_OXA-40_ gene was detected. The carbapenemases genes/classes were correctly determined in the remaining 57 (98.3%) isolates (see Figure 1 and Figure 2 for the conventional PCR and Figure 3 and Figure 4 for the real-time PCR results).

The time required to obtain the culture was the same for all the methods, but a further detection time varied significantly with the methods tested: the application of phenotypic methods required approximately 2 h, 18 h and 20 h for CarbAcineto NP, CPO and CIM, respectively. The real-time PCR results were obtained after approximately 2 h, while by conventional PCR after 3 h.

## 3. Discussion

Infections caused by *A. baumannii*, most often resulting from MDR strains, are a global threat and cause serious treatment difficulties [2,13,14]. Carbapenems are usually the drugs of choice in the treatment of *A. baumannii* infections. However, the high plasticity of the *A. baumannii* genome allows for the easy acquisition of resistance genes, as well as those encoding enzymes that hydrolyze carbapenems [15]. The scientific literature underlines that the most common acquired carbapenem resistance mechanisms among *A. baumannii* isolates are class D beta-lactamases [13,16], which is in accordance with the results of our study (all of the strains synthesized OXA-like enzymes).

It is of great importance to trace the spread and dissemination of CHDLs [17,18]. As there is no selective inhibitor for CHDL and there is no possibility of a selective phenotypic determination of these carbapenemases producers among *A. baumannii* isolates, different methods are used to detect the mechanisms of enzymatic resistance to carbapenems. In this study, five methods of CHDL detection were evaluated: CarbAcineto NP, CIM, panels compatible with the BD Phoenix™ M50 system (CPO Panels, Becton Dickinson, Franklin Lakes, NJ, USA), conventional Polymerase Chain Reaction and real-time PCR. The IVD-certified eazyplex^®^ SuperBug complete A assay (AmplexDiagnostics GmbH, Gars am Inn, Germany), which allows for the detection of the most important beta-lactamases genes amongst Gram-negative rods, was used as a reference method.

CarbAcineto NP is a phenotypic test that detects the hydrolysis of carbapenem (imipenem), so it does not specify a type of carbapenemase [19]. In the present study, the sensitivity of this method, compared with the reference test, reached 65.5% (*n* = 38). Dortet et al., in turn, showed a high sensitivity and specificity of this method: 94.7% and 100%, respectively [19]. In other studies performed in Poland among *Acinetobacter* spp. (*n* = 58), the isolates’ sensitivity to CarbAcineto NP was 88.9%. In total, 46 (79.3%) of the strains in the mentioned study were CHDL producers [20]. In the present study, the corresponding value reached 100%. Taking into account that oxacillinases usually present a weak hydrolytic activity of carbapenems, even a slight color change should be considered as a positive result. It is unlikely for the CarbaNP results obtained for other Gram-negative rods, for which the color change is usually more significant. The highest advantage of this method is relatively the shortest time to obtain results, compared to the remaining tested methods. However, an interpretation of the CarbAcineto NP test results is sometimes quite difficult.

CIM is another phenotypic method, for which, in the present study, positive results were obtained for all (*n* = 58) of the examined strains. It showed a higher sensitivity than the CarbAcineto NP test, possibly due to the clear, simple and objective results interpretation criteria. CIM results are based on the measurement of the size of the bacterial growth inhibition zones. Results similar to ours were also obtained in another study, in which the sensitivity of CIM was 88.1% (*n* = 37) among CHDL-positive *A. baumannii* strains (*n* = 42) [21]. The opposite results were obtained by Simner et al., in which the sensitivity of the CIM test was 29.0%, but for the mCIM version, the corresponding value was 71.0% [22]. These results arose from the incubation time of the meropenem disc (CIM-2 h, mCIM-4 h) in bacterial suspensions. Thus, in our study, the incubation time was also extended to 4 h to increase the sensitivity of the results.

For a better understanding of phenotypic methods, it should be noted that CHDLs are synthesized at a low level by bacteria which do not provide the efficient carbapenem hydrolytic effect [23]. A particular advantage of phenotypic methods is the ability to determine the actual presence of carbapenemase and confirm in vitro the hydrolysis of carbapenem by the synthesized enzyme. 

For all of the studied isolates, carbapenemases were detected using CPO panels (sensitivity 100%, *n* = 58); additionally, for 91.4% of the strains (*n* = 53), the beta-lactamase class was also determined correctly. These results were similar to those obtained in previous studies. Whitley et al. showed that out of 70 carbapenemase-producing *A. baumannii*, 91.7% were true positive [24]. Other studies also showed a sensitivity of 100% for the detection of carbapenemases with CPO panels, while 88.6% of the AST profiles were correctly classified to a particular beta-lactamases class [25]. An advantage of CPO panels is the simultaneous determination of antimicrobial susceptibility and automatic classification of beta-lactamases class. This allows for a faster detection of a particular resistance mechanism and choice of the appropriate therapeutic option at an earlier stage of the infection.

Since the cost of methods used in routine practice is one of the most important factors in selecting tests for use in a particular laboratory, a comparison of the compared methods was also summarized. CarbAcineto NP, CIM, the conventional PCR and real-time PCR were relatively inexpensive (around 1–2 EUR per strain, using the mentioned reagents). CPO panels were the most expensive, with a cost of around 6 EUR per strain (all calculations excluded the cost of the reference strains).

The detection time was another important factor of the chosen laboratory procedure. Results could be obtained in the shortest and comparable time using CarbAcineto NP, conventional and real-time PCR. The detection time for CIM and CPO panels was approximately 18–20 h, but in the CPO panel, the detection of carbapenemases was performed simultaneously with the antimicrobial susceptibility testing.

One of the biggest limitations of the study was that when describing or comparing the efficacy of the methods of carbapenemases synthesis, carbapenemase-negative isolates should have been included, as well as positive ones. We only applied the carbapenemase-negative reference strain as quality control in each method. The second was that we only investigated a limited number of 58 strains, because our goal was to use the uniform group of strains with similar susceptibility profiles and with the confirmed synthesis of oxacillinases. In our region, as well as in our unit, OXA-23 and OXA-40 producers are the most often isolated from clinical material. Moreover, this study was planned for the reliable detection of CHDL enzymes. Therefore, the strains with other resistance mechanisms were neither available nor the main topic of the study.

This study demonstrated that CHDL detection using molecular biology-based methods is the most reliable. The results obtained by the conventional PCR and real-time PCR showed the highest sensitivity in the determination of a carbapenemase type. In one isolate, another *bla*_OXA_ gene was detected, which may indicate the heterogeneity of a given strain, coculture or a simple laboratory mistake. Studies by other authors also show a high sensitivity of the PCR (100%) applied for this particular purpose [26]. We preceded the real-time PCR by thermal DNA isolation. Thus, using our approach for the detection of the carbapenemases gene, results could be obtained in an even shorter amount of time (up to 2 h for the laboratories working around the clock). The difference in time to obtain the results between the conventional PCR (3 h) and real-time PCR (2 h) resulted from the necessity for the electrophoretic separation of amplicons to visualize the results in conventional PCR. Nevertheless, molecular methods are probably the most reliable and the fastest tools to detect carbapenemases. This is of particular clinical and epidemiological importance.

## 4. Conclusions

OXA-like enzymes were the most prevalent carbapenemases amongst *A. baumannii* isolates, but could not be precisely and easily determined with the application of phenotypic methods. In contrast, only phenotypic methods allowed the determination of the actual in vitro hydrolysis of an antibiotic. However, their sensitivity may be lower due to a decreased enzyme synthesis level. The CPO panels presented the ability to evaluate the susceptibility of the strains to several antimicrobial agents simultaneously in one panel as routine procedure and, additionally, to detect carbapenemases types at the same time. In turn, molecular methods allowed for the determination of a specific gene encoding CHDLs and presented the highest sensitivity and specificity for that particular purpose. In summary, the compared methods are applicable for the detection of CHDLs, but require an experienced investigator to obtain an appropriate read out of the results (e.g., CarbAcineto NP test). However, the sensitivity of the investigated methods may be lower and the carbapenemases assignation may be less reliable when compared to molecular biology-based methods.

## 5. Materials and Methods

### 5.1. Bacterial Strains and Their Origin

This study involved 58 *A. baumannii* strains, each isolated from a different patient of Antoni Jurasz University Hospital No. 1 in Bydgoszcz, Poland, between 2017 and 2019. In total, 40 (69.0%) were derived from the patients of the Anesthesiology and Intensive Care Unit (for the detailed origin of the strains, see Appendix A).

The examined strains were isolated from various clinical specimens. The majority, 26 (44.8%), of *A. baumannii* strains were derived from bronchoalveolar lavage samples, and 13 (22.4%) from wound swabs (Appendix A).

*A. baumannii* DSMZ (Deutsche Sammlung von Mikroorganismen und Zellkulturen, Germany) 30008 (carbapenemase-negative control strain) and *A. baumannii* DSMZ 102930 (carbapenemase-positive control for phenotypic tests) were used as the quality control strains for the applied methods. *Escherichia coli* ATCC (American Type Culture Collection) 25922 was applied for CIM test as an indicator strain.

### 5.2. Antimicrobial Susceptibility

Antimicrobial susceptibility tests (AST) were determined with BD Phoenix™ M50 NMIC-402 panels (Becton Dickinson, Franklin Lakes, NJ, USA) in a routine laboratory diagnostic scheme. The results of AST were interpreted according to European Committee on Antimicrobial Susceptibility Testing recommendation v 10.0 (EUCAST, 2020) [27].

### 5.3. Carbapenemases and Carbapenemase Genes Detection

Five methods (CarbAcineto NP, Carbapenem Inactivation Method, BD Phoenix™ M50 with GN-502 panels, conventional PCR and real-time PCR) were performed to evaluate carbapenemases or their gene presence among the studied strains.

### 5.4. Phenotypic Tests

CarbAcineto NP: a full inoculation loop (10 μL) of the overnight culture of the examined strain was collected from Columbia Agar with 5% Sheep Blood (CAB, Becton Dickinson, Franklin Lakes, NJ, USA) plates and suspended in two 1.5 mL tubes (named A and B) each containing 100 μL of 5 M NaCl and pH indicator (phenol red). B tube was supplemented with imipenem at the final concentration of 6 mg/mL (Fresenius Kabi, Bad Homburg, Germany)**.** Both solutions were then incubated at 37 °C for a maximum of 2 h. An optical read-out of the color change of each solution was conducted afterwards. Carbapenemase activity was detected by a color change (red to orange/yellow) of phenol red in the B tube and the absence of color changes in the A tube [19].

CIM: a full inoculation loop (10 μL) of the overnight culture of the tested strain was suspended in 1.5 mL tube containing 400 μL of a distilled water. The meropenem disc (10 μg) (Oxoid Ltd., Basingstoke, UK) was added afterwards. The suspension with meropenem disc was incubated for 4 h at 35 °C. After the incubation step, the meropenem disc was removed from suspension and placed on Mueller–Hinton agar (MHA, Becton Dickinson, Franklin Lakes, NJ, USA) plate inoculated with carbapenem-susceptible *E. coli* ATCC 25922. The culture was incubated for 18 h at 35 °C. The CIM results were interpreted based on the inhibition zone diameter size and considered positive if ≤17 mm, indeterminate if 18–19 mm, and negative if ≥20 mm [28].

BD Phoenix™ M50 with GN-502 panels (Becton Dickinson, Franklin Lakes, NJ, USA): AST indicator and 25 µL of bacterial suspension prepared initially in the dedicated solution (both provided by a manufacturer, BD Phoenix™ ID Broth) were added into Mueller–Hinton broth (MHB, Becton Dickinson, Franklin Lakes, NJ, USA). The panels were inoculated with these suspensions subsequently and placed in BD Phoenix™ M50 device. The results with their interpretation were obtained automatically within 18 h.

### 5.5. Molecular Biology-Based Approach

#### 5.5.1. DNA Template Extraction

A loopful (1 μL) of an overnight culture of bacteria was added to 300 µL of pure molecular biology-grade water (EurX, Gdańsk, Poland) and boiled for 10 min in 100 °C in 1.5 mL tube. The bacterial suspension was then vortexed and centrifuged at 12,800 rpm for 1 min. The supernatant was used for the reaction as DNA template, as previously described by Abrar et al. [29].

#### 5.5.2. Conventional “In House” PCR for the Detection of *bla*_OXA_ Genes

PCR assay was performed using the primers described previously by Woodford et al. [30] for *bla*_OXA-40_ F: 5′-GGTTAGTTGGCCCCCTTAAA—3′; R: 5′-AGTTGAGCGAAAAGGGGATT—3′ (product size—246 bp); for *bla*_OXA-23_ F: 5′-GATCGGATTGGAGAACCAGA—3′; R: 5′-ATTTCTGACCGCATTTCCAT—3′ (product size—501 bp) (both manufactured by Genomed, Warsaw, Poland). The final volume of the PCR was 25 µL. The following reagents (all Promega, Walldorf, Germany) were used: MgCl_2_—2.5 mM; dNTPs—1 mM; polymerase GoTaq—1 U/reaction; primers—200 pM, 1.5 × concentrated polymerase buffer and 2 µL of a DNA template. An amplification of each target gene was performed using a GeneAmp^®^ PCR System 2700 (Applied Biosystems, Foster City, CA, USA). Both reactions were performed according to the following conditions: 1 cycle at 94 °C for 5 min, followed by 30 cycles at 94 °C for 25 s, 52 °C for 40 s, and 72 °C for 50 s with a final elongation at 72 °C for 6 min. The PCR reaction products were separated by electrophoresis on 1.5% agarose gel (Bio-Rad, Feldkirchen, Germany) in a 1 × Tris-Boric Acid-EDTA (TBE, Bio-Rad, Feldkirchen, Germany) running buffer at 9 V/cm for 1 h in a MINI SUB^TM^ DNA CELL (Bio-Rad, Feldkirchen, Germany) device.

Real-time PCR was performed using the same primers. The final volume of the reaction was 20 µL. Reagents per reaction were used as follows: 4 µL of 5 × HOT FIREPOL^®^ EvaGreen^®^ (SolisBiodyne, Tartu, Estonia), 5 µL of molecular biology grade water (EurX, Gdańsk, Poland), 5 µL of each primers (0.25 µM/reaction) and 1 µL of a DNA template. The following conditions were applied: 1 cycle at 95 °C for 12 min, followed by 40 cycles at 95 °C for 15 s, 58 °C for 20 s, and the final elongation—72 °C for 20 s. Data collection was enabled at each extension step. The melt curve protocol was added afterwards—95 °C for 5 s and 60 °C for 1 min—, followed by data acquisition at 0.11 °C increments between 60 °C and 97 °C to confirm the specificity of the amplification product (for the results, see Appendix A). Data collection was enabled continuously at each increment during the temperature change (high-resolution melting). Reactions were performed on cobas z480 analyzer (Roche, Basel, Switzerland).

eazyplex^®^ SuperBug complete A assay (AmplexDiagnostics GmbH, Gars am Inn, Germany), performed according to manufacturers’ instruction, was used as a reference method.

### 5.6. Data Analysis

The data were analyzed by comparison of the results with the reference assay. The results were assessed in three categories: detection of enzyme/carbapenemase, assignation to a carbapenemase class and differentiation to the type of carbapenemase. Statistical analysis of the results was performed using chi-square test.

## Figures and Tables

**Figure 1 antibiotics-11-00455-f001:**
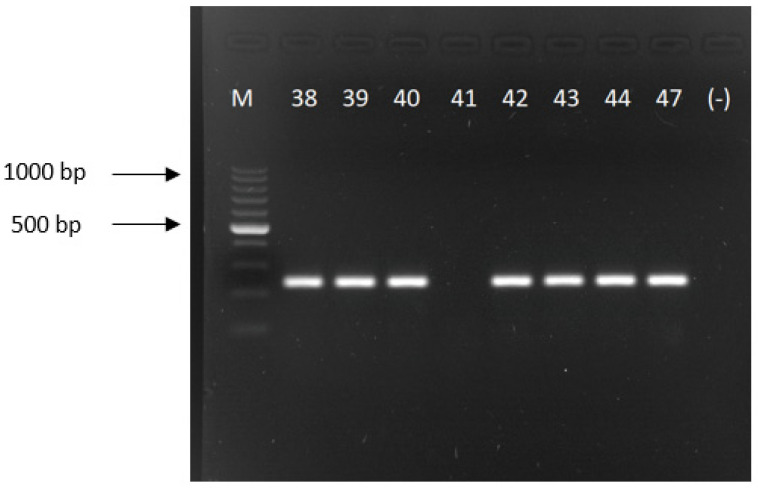
Picture of an electrophoretic gel showing the PCR amplicons for *bla*_OXA-40_ gene; product size of 246 bp (M—DNA size marker 100–1000 bp; 38-44—numbers of the examined strain positive for *bla*_OXA-40_ gene except for number 41; 47—positive control; (-)—negative PCR control).

**Figure 2 antibiotics-11-00455-f002:**
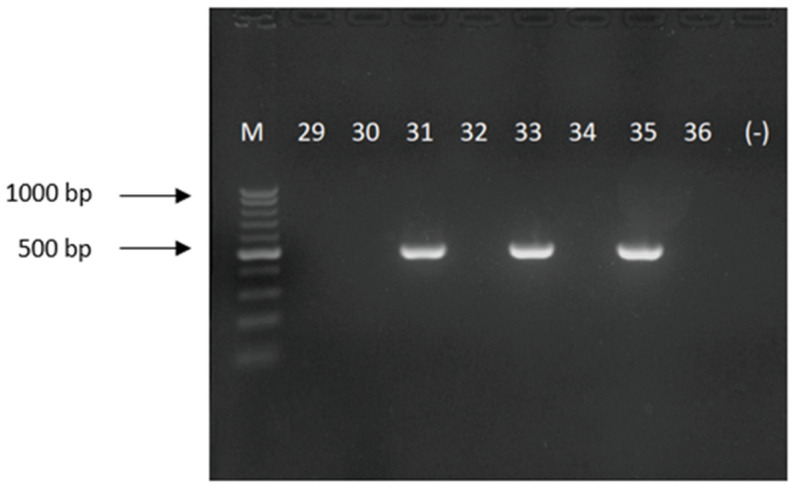
Picture of an electrophoretic gel showing the PCR amplicons for *bla*_OXA-23_ gene; product size of 501 bp (M—DNA size marker 100–1000 bp; 29-36—numbers of the examined strain, with 31, 33 and 35 positive for *bla*_OXA-23_ gene; (-)—negative PCR control).

**Figure 3 antibiotics-11-00455-f003:**
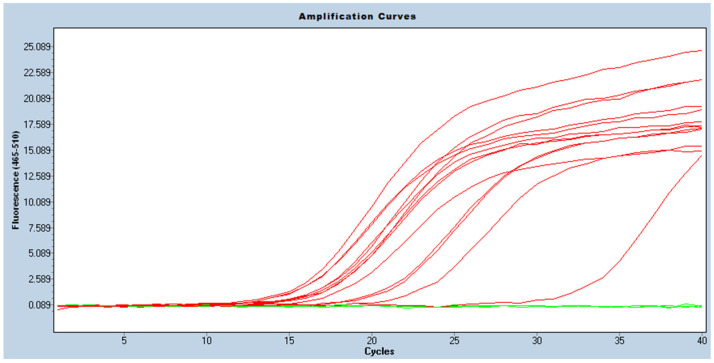
Picture of amplification curves showing results of real-time PCR for *bla*_OXA-40_ gene (red curves—positive results; green curves—negative results).

**Figure 4 antibiotics-11-00455-f004:**
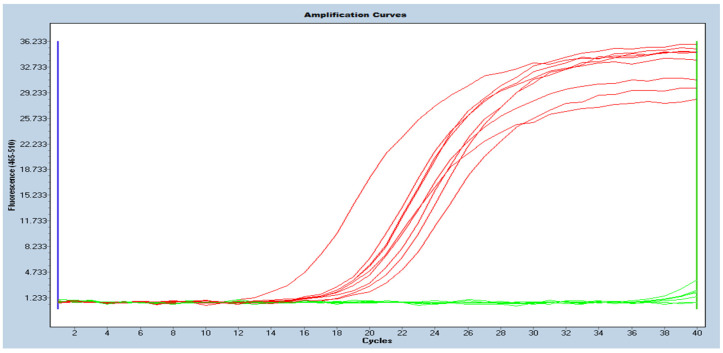
Picture of amplification curves showing results of real-time PCR for *bla*_OXA-23_ gene (red curves—positive results; green curves—negative results).

**Table 1 antibiotics-11-00455-t001:** Antimicrobial susceptibility profiles of the tested isolates (*n* = 58).

Antimicrobial	Resistant Strains (*n* = 58) Number (%)
Imipenem	58 (100%)
Meropenem	58 (100%)
Gentamicin	43 (74.1%)
Amikacin	55 (94.8%)
Tobramycin	56 (96.6%)
Ciprofloxacin	58 (100%)
Levofloxacin	58 (100%)
Trimethoprim/sulfamethoxazole	57 (98.3%)
Colistin	5 (8.6%)

**Table 2 antibiotics-11-00455-t002:** Detection of a particular carbapenemase class by phenotypic and genotypic methods.

	No. of Isolates Positive for a Particular Class of Beta-Lactamases/Strains Number (%)
Assay	Class D carbapenemase
CPO	53/58 (91.4%)
Real-time PCR	58/58 (100%)
Conventional PCR	58/58 (100%)

**Table 3 antibiotics-11-00455-t003:** Number of isolates producing carbapenemases detected by the tested methods compared to the reference method.

Assay	No. of Positive Results of Carbapenemase Detection/Strains Number (%)
CarbAcineto NP	38/58 (65.5%) *
CIM	58/58 (100%)
CPO	58/58 (100%)
Real-time PCR	58/58 (100%)
Conventional PCR	58/58 (100%)

* statistically significant difference *p* > 0.05.

## Data Availability

The data presented in this study are available on request from the corresponding author.

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
