# Peer review of "Conventional and Real-Time PCR Targeting blaOXA Genes as Reliable Methods for a Rapid Detection of Carbapenem-Resistant Acinetobacter baumannii Clinical Strains"

_antibiotics, 2022, doi:10.3390/antibiotics11040455_

Round 1

Reviewer 1 Report

The authors presented a study which would be useful in determining a confirmatory test for Acinetobacter. Although not pointed out in the conclusions mCIM performed very well when compared to commercial phenotypic tests. Actually this result is important for countries and labs with limited resources.

However the study has several limitations;

1-First of all the number of isolates are very limited.

2-As I have remarked on the draft also the panel should have included isolates with different enzyme types (CHDLs (acquired and intrinsic) ) and also isolates /strains negative for acquired CHDLs ( as intrinsic ones would be positive anyways) and which contains other enzymes (ESBLs like PER, GES etc)

3- Although getting the results rapidly is  important as it was even cited in the title; there is no mention about the time to get the results in the Results section.

4- The “gold standart” is a commercial test and the authors did not give any references about the performance of the test so I take it as an  arbitrary reference . They should give evidence that this test is the equivalent of sequencing. This is commercial and special instruments are needed.

5-For one isolate discrepant results were get by both of the PCR assays and the “reference” test. The identity of this OXA should be confirmed by sequencing.

6- Cost effectiveness is very important for routine laboratories. So it should also be mentioned in the Discussion.

7-  The paper should be read and corrected by a native English speaker                                                                                                                                       

Other remarks and questions are in sticky notes on the draft.

Reviewer 2 Report

The manuscript of D. Depka et al. titled "Real-time and standard PCR targeting blaOXA genes as a reliable methods for a rapid detection of carbapenem-resistant Acinetobacter baumannii clinical strains" devoted to the detection of blaOXA genes in clinical strains of Acinetobacter baumannii by comparison of phenotypical and molecular methods, namely cPCR and qPCR. The manuscript is well-written and didn't find major issues.

The minor issues and suggestions are given below:

1) I suggest considering using terms quantitative and qualitative PCR in the title and throughout the manuscript instead of real-time and standard ones. Also, the authors can use the term "conventional" instead of "standard" for PCR to avoid ambiguity.

2) Omit "a" before "reliable methods" in the title.

2) Line 6: Add "*" for the corresponding author.

3) Line 202: What are the five methods?

4) I suggest placing results mentioned at lines 251 and 263 in the Results section.
